# Adolescent awareness and experience of the pubertal changes: A qualitative study from Rwanda

Kellen Muganwa[1]*, Oliva Bazirete[1], Marie Chantal Uwimana[1],
Marie Grace Sandra Musabwasoni[1], Olive Tengera[1], Joy Bahumura[1],
Hellen Nwanko Chinwe[1], Emmerence Uwingabire[1], Elyse Muteteri,
Francoise Mujawamariya[1], Emmanuel Habyarimana[1], Eugenie Mukeshimana[1],
Baudouine Ndimurukundo[1], Verene Abagirinana[1], Richard Nsengiyumva[1],
Christine Mbila Wabenya[1], Vincent Mugisha[2], Gerard Kaberuka[3],
Marie Laetitia Ishimwe Bazakare[3], Thierry Claudien Uhawenimana[1]

**1** School of Nursing and Midwifery, College of Medicine and Health Sciences, University of Rwanda, Kigali, Rwanda, **2** University of Teaching Hospital, Kigali, Rwanda, **3** University of Rwanda, Single Project Unit, Kigali, Rwanda

* muganwakellen@gmail.com

## Abstract

### Introduction

The transition from puberty to adolescence involves significant body changes that manifest differently across young children. It is essential that adolescents become aware of these changes so that they can easily navigate puberty's physical, emotional, and interpersonal changes safely in terms of sexual and reproductive health. In Rwanda, there remains a gap in literature on adolescents' awareness and experiences of pubertal changes among school going adolescents; particularly among those enrolled as day students. This research was undertaken to explore the awareness and understanding of puberty-related changes among school-aged adolescents, as well as their personal experiences of these changes. The findings aim to enhance the sexual education curriculum in Rwanda, with a particular emphasis on day students in rural regions who may be vulnerable to peer and societal pressures that can potentially expose them risky sexual behaviours.

### Methods

A qualitative study was conducted with 120 school-going adolescent girls and boys. Participants were selected using a purposive sampling approach. Twelve focused group discussions were formed in six secondary schools from rural and urban 12-years basic education in the Eastern province of Rwanda. Deductive thematic analysis was used to analyse data using Dedoose software.

**Data availability statement:** Kinyarwanda transcripts that have informed the drafting of this paper are provided as supporting information.

**Funding:** The author(s) received no specific funding for this work.

**Competing interests:** The authors have declared that no competing interests exist.

## Results

The analysis yielded two main themes: i) what adolescents know about the changes occurring during puberty, encompassing physical, emotional, and behavioral transformations, as well as the social interactions linked to these changes and their perceptions regarding the factors that influence these changes, and ii) personal experiences pertaining to the changes associated with puberty. Most of the participants were aware of body changes that happen during puberty. While puberty marked an achievement of maturity for most of the participants, the onset of menarche for female adolescents was unique due to the embarrassment it caused them.

## Conclusion

The study found that some young girls experience menarche unprepared and get inadequate support to initially navigate that change. Therefore, caregivers should proactively get educated on pubertal changes so that they provide appropriate support to children to navigate puberty safely.

## Introduction

Puberty is a period characterized by significant changes in physical, neurological, behavioral, cognitive, and emotional domains [1,2]. It marks a transition between childhood and adulthood, encompassing a period of rapid growth and development across physical, psychological, and social dimensions [3,4]. Research indicates that when puberty is not navigated appropriately, it may be a critical period during which vulnerability to sexual risky behaviours heightens [5,6]. This disruption can result in various psychological issues, such as anxiety, depression, eating disorders, and addictive behaviors [5,6]. The onset of puberty occurs concurrently with the school years for many children, leading to intensified feelings of shame and heightened sensitivity [7–9]. This is largely attributable to a range of physiological changes, including menstruation in girls, as well as adverse peer influences, and challenges related to body image [9, 10]. Compounding these issues is the fact that some young children may often navigate these changes with minimal support. This can hamper their ability to effectively cope with the changes associated with puberty throughout their next developmental stages thereby affecting their sexual and reproductive health largely.

Research indicates that young children with information about puberty are better positioned to deal with challenges associated with this developmental stage [9,11–14]. However, discussion about pubertal matters remains taboo in the context of low income settings [15–17]. Therefore, it is important to research about this topic in order to find the safest ways of providing children with trusted and accurate information about pubertal changes and sexual and reproductive health matters associated with them [18,19]. Most importantly, obtaining information about adolescents' awareness and experiences of pubertal changes can inform the design of educational interventions to assist young individuals in dispelling common sexual and

reproductive health misconceptions, thereby mitigating potential negative outcomes such as teenage pregnancies, sexually transmitted infections, and substance abuse [18].

Like in other low income settings, in Rwanda, cultural stereotypes are still affecting the provision of sexuality education [4,20]. Though adolescents undertake a course on sexual and reproductive health at school, it is not sufficient for them to get all essential knowledge needed to navigate their puberty safely [21]. This is evidenced by reports from the media indicating that in 2021, between 2016 and 2018, some 55,048 girls below 18 years were impregnated in the whole country with the Eastern Province being most affected with 19,838 cases. Research conducted in Rwanda identified, among various factors, the insufficient awareness of sexual and reproductive health among adolescents as a significant contributor to the increase in teenage pregnancies [17,22].

In societies such as Rwanda, especially in rural regions where discussions about sexuality are often considered taboo, children may suffer from a lack of knowledge regarding their bodily and emotional transformations. This ignorance can result in adverse outcomes, including the spread of sexually transmitted infections and unintended pregnancies among girls, which may force them into early parental roles, lead to school dropouts, and produce other negative consequences. It is crucial to gather information on what young individuals understand about the changes that occur during puberty, as well as their personal experiences during this developmental phase. Such insights are essential for enhancing existing sexual education programs for youth by identifying gaps in adolescents' awareness and understanding of pubertal and reproductive changes, as well as promoting their ability to maintain good health during this critical period. When children enter secondary school with conflicting information regarding their physical and emotional transformations, coupled with peer pressure to engage in early sexual risk behaviors, they face significant challenges. Their developmental immaturity exacerbates their susceptibility to exploitation by individuals who may take advantage of their innocence and limited understanding of how these bodily changes can lead to sexual exploitation and its negative consequences. In the context of rural Rwanda, existing research has not sufficiently addressed adolescents' comprehension of puberty and their personal experiences during this critical phase of development. This gap in knowledge creates uncertainty regarding how adolescents can utilize their understanding of pubertal changes to navigate their sexuality during the early stages of their school years. This is particularly relevant in rural areas of Rwanda, where day school children may lack adequate support networks to mitigate the adverse effects of insufficient knowledge about puberty and appropriate behaviors during this transitional period. Therefore, this study aimed at exploring adolescents' awareness and experiences of pubertal changes in order to garner insights that can guide the design of tailor-made interventions addressing gaps in the sexual education about pubertal changes in Rwandan schools [17,23].

### Objectives of the study

- To explore if adolescents in secondary schools in the Eastern Province of Rwanda are aware of bodily and emotional changes occurring during puberty

- To understand from adolescents' perspectives how they experience pubertal changes and sources of support during this period.

### Methodology

The reported findings are from the initial phase of a qualitative study that focused on garnering young adolescents' understanding of the physiological and emotional changes taking place during puberty. The second phase of the study focused on how the pubertal changes can affect young adolescents' sexuality and reproductive health. A descriptive qualitative design was used to explore adolescent children's awareness of physiological and psychological changes occurring during puberty, and how they experienced these changes. The phenomenon of pubertal changes in Rwanda remains

underexplored in academic literature. Therefore, the selection of a descriptive qualitative design for this study is justified, as the research question aims to elicit insights regarding the concept of pubertal changes from the viewpoints of children currently experiencing puberty.

## Setting

The study was carried out in six secondary schools selected in rural and urban areas located in three districts that is Rwamagana, Kayonza, and Gatsibo districts in twelve years basic education as Rwanda education program in grassroot level. The Eastern Province was chosen because of high prevalence of teenage pregnancies in Rwanda [24].

## Study population

The targeted population of this study were adolescent children who were studying in twelve years basic education [25]. In Rwanda, twelve-year basic education denotes an educational framework that guarantees at least twelve years of formal schooling, generally covering both primary and secondary education stages. This system consists of six years of primary education, followed by three years of junior secondary education (O-Levels), and concludes with three years of senior secondary education (A-Levels). Participants in the study were required to be between the ages of 10 and 19 and to be enrolled in grades senior 1 through senior 6. The research encompassed both young females and young males who were attending twelve-year secondary schools as day students. They had to be living in rural and urban areas of Rwamagana, Kayonza and Gatsibo districts in the Eastern Province of Rwanda during their study period. Adolescents whose guardians did not provide consent for participation were excluded from the study. Additionally, those within the same age group who had given birth and were enrolled during the designated academic years were also omitted, as their experiences of puberty may substantially differ from those of their peers who have not yet entered parenthood. Furthermore, young girls and boys attending senior 1 to senior 6 who were enrolled as boarding school students were excluded from the study. This decision was based on the assumption that they possess protective factors against vulnerabilities to sexual risk behaviors, in contrast to their peers who were enrolled in day schools within the study settings.

## Sample size and sampling approach

We used a purposive sampling approach to select six secondary schools. Selection criteria varied in terms of rural and urban settings, as well as the owners of the schools to ensure representativity. For each selected institution, we conducted focus group discussions exclusively with female and male students, resulting in a total of twelve focus group discussions (FGDs).

To ascertain the number of participants for these discussions, we employed a maximum variation purposive sampling method to ensure representative of the schools where the study took place. The criteria for children's eligibility to participate were established according to the inclusion parameters defined for the study. In collaboration with teachers from the respective classes, we randomly selected two students from each class, ranging from senior 1 to senior 6, as these students had not yet reached the age of 18 during this period. This resulted in 120 participants from three districts selected for this study.

## Data collection

The recruitment of participants and the data collection exercise for this study took place between 10th August 2022 and 8th December 2022. A topic guide was created in alignment with the study's objectives and the current literature about pubertal changes and the experiences of adolescents undergoing these changes. This covered three areas: i) the physical, emotional, and social changes occurring during puberty, ii) views about the possible causes of these changes, and iii) the lived experiences of adolescents during this development phase. Additional probing was conducted based on participants' responses to elicit more comprehensive information regarding their answers.

The original topic guide was composed in English but was subsequently translated into Kinyarwanda, the national language of Rwanda, to enable young boys and girls participating in the study to share their insights regarding the research focus without any barriers.

During the data collection phase, focus group discussions were conducted to gather insights from adolescent boys and girls regarding the changes associated with puberty. To mitigate power dynamics and gender disparities among participants, the focus groups were organized by gender. Separate sessions were held exclusively for male participants and for female participants. Distinct discussion rooms were established to maintain confidentiality, thereby fostering a comfortable environment that encouraged each gender to share their knowledge and experiences related to puberty more openly.

The discussion lasted approximately 45–60 minutes. All discussions were recorded after getting permission from the participants to avoid distraction that may result in focusing on note taking during the interview. In addition, field notes and memos were taken to comment and maintain researcher's impressions, thoughts, and feelings about environment contexts, adolescent's non-verbal expressions that were not sufficiently captured through recorded interviews [26].

## Data management and analysis

Collected data were transcribed verbatim by the research team members. The interviews that were gathered were initially transcribed in Kinyarwanda and subsequently translated into English by members of the research team who possessed proficiency in both languages. Deductive thematic analysis was employed to examine the data [27,28]. The themes were established in advance, aligned with the study's objectives, and included: i) adolescents' understanding of pubertal changes, ii) personal experiences related to pubertal changes. Transcripts were first read and re-read to familiarise with data (see S1 File. Kinyarwanda transcripts-Girls and S2 File. Kinyarwanda transcripts-Boys). The next stage involved the coding process and generation of sub-codes. Relevant words, phrases, and segments from participants' accounts were identified to reflect the subcodes of each predetermined theme. Whenever appropriate, these subcodes led to the development of sub-themes corresponding to each overarching theme. Dedoose software was used to manage and organize transcribed data to enhance the process of data analysis.

## Strategies undertaken to ensure trustworthiness

To ensure credibility of the study, researchers remained truthful to the adolescents' perceptions of their sexual and reproductive health changes. The tenet of credibility was established through a purposive selection of the participants who met the criteria and were able to share their experiences in regard to the perceptions of their physiological changes. The researchers discussed the outcomes from experiences regarding the study phenomena and came up with a common consensus about the description of findings that were generated from participants' group discussions. Transferability was enhanced through a deep description of the study methods that allow the future researcher to employ the same methods and conduct the same study in different environments.

## Ethical considerations

The study was approved by the Institutional Review Board (IRB) of University of Rwanda, College of Medicine and Health Sciences, approval notice: Nº: 303/CMHS IRB/2022 (see supplementary information 1). Additional permissions to start data collection were obtained from respective mayors of the districts where the study took place. Written informed consent and assent was requested from parents/ guardians of research participants prior to the participation of the study. By signing the informed consent and assent, participants gave consent for interview transcripts to be published. Participants were assigned codes to ensure their confidentiality and anonymity. The participants also have the right to refuse, participate or to withdraw from the study at any time without any punishment or consequences whatsoever.

## Results

In total, twelve focus group discussions including six FGD involving adolescent male students and six FGD involving adolescent female students were conducted across six secondary schools(three schools in rural and three more in urban areas). In total, the study involved 118 participants including 58 boys and 60 girls. Other sample characteristics are reported in Table 1 below:

As shown in the above table, the mean age of participants was 16.6. The majority of participants (91/118) were doing their ordinary level studies. Over half of participants (57.6%) stayed with their caregivers in the village after school and 61% (n = 72) spent their holidays in the villages. As shown above, 51 participants were not living with both parents during the period of data collection.

### Themes that emerged from the analysis

Two themes and their sub-themes are summarized in Table 2 below.

**Table1. Participants demographic data.**

| Variable | | N | % |
|---|---|---|---|
| **Age in years** | 13.00 | 5 | 4.2 |
| | 14.00 | 19 | 16.1 |
| | 15.00 | 14 | 11.9 |
| | 16.00 | 30 | 25.4 |
| | 17.00 | 26 | 22.0 |
| | 18.00 | 24 | 20.3 |
| | Total | 118 | 100.0 |
| **Year of study** | 1.00 | 39 | 33.1 |
| | 2.00 | 33 | 28.0 |
| | 3.00 | 19 | 16.1 |
| | 4.00 | 9 | 7.6 |
| | 5.00 | 13 | 11.0 |
| | 6.00 | 5 | 4.2 |
| | Total | 118 | 100.0 |
| **Residence during the class period** | Village | 68 | 57.6 |
| | Town | 50 | 42.4 |
| | Total | 118 | 100.0 |
| **Residence during holidays** | Village | 72 | 61.0 |
| | Town | 46 | 39.0 |
| | Total | 118 | 100.0 |
| **Caregiver's profile** | Both parents | 67 | 56.8 |
| | Only lives with the mother | 46 | 39.0 |
| | Only lives with the father | 2 | 1.7 |
| | Lives with relatives | 3 | 2.5 |
| | Total | 118 | 100.0 |
| **Religion** | Christians | 102 | 86.4 |
| | Islam | 16 | 13.6 |
| | Total | 118 | 100 |

**Table 2. Themes that emerged from the analysis.**

| Major themes | Subthemes |
|---|---|
| Adolescents' understanding of pubertal changes | Bodily physical and physiological changes |
| | Emotional and behavioural changes |
| | Social interaction changes |
| | Causes of changes boys and girls experience during puberty |
| Personal experiences related to pubertal changes | Though pubertal changes may lead to panic, they embody the attainment of maturity among young boys |
| | Pubertal changes affect young girls' perception of their body image |
| | Menarche causes panic and embarrassment to young girls |
| | Lack of parental support for some young girls |

### Theme One: Adolescents' understanding of pubertal changes

This theme covers four sub-themes covering adolescents' awareness of bodily and emotional changes that occur during puberty and causes of these changes.

**Subtheme 1: Bodily physical and physiological changes.** Both young girls and boys were capable of mentioning more than one of the physiological changes that young children notice when they reach puberty. The majority of the participants mentioned that when girls reach puberty, their body shape changes and particularly their hips widen, and their buttocks get curvier. They also mentioned that girls develop breast budding and growing tenderly. Both young girls and boys mentioned other physical changes associated with puberty for both sexes including hair growth around the pubic area and underarms. Another important change mentioned was that when girls reach puberty, they see their first menarche. They also mentioned that some girls during puberty may develop acne.

As for body changes young boys see during puberty, the majority of the participating young girls added that boys experience frequent nocturnal emissions. They also mentioned that boys grow hair on their chest, and other parts of the body such as arms and legs. Young girls stated other important body changes such as the widening of the chest and muscles, deepening of the voice, and rapid increase in the body size and growth that boys develop. They also mentioned that boys too may get acne resulting from the pubertal changes. A few young girls mentioned that during puberty, boys develop an increased appetite.

**Subtheme 2: Emotional and behavioural changes.** Young boys mentioned some the emotional and behavioral changes they observed on their female counterparts. The positive changes mentioned included caring for their physical appearance, increased self-respect, feeling a sense of responsibility, heading to their parents' advice, and helping parents at home. Young boys reported that their female counterparts start having an urge of having boyfriends whom they may constantly stay together. Young girls confirmed some of the psychological changes mentioned by male participants and added many more changes that girls experience during adolescence. The majority of young girls reported that when girls reach puberty, they experience mood swings. They revealed that during puberty some young girls get preoccupied with their body image and change the way they walk, particularly in public places.

Like their male counterparts, young girls reported that they too can develop an urge of having a friend of the opposite sex. Some young girls reported that they may develop uncontrolled lust.

"*I've heard most of my colleagues saying that when they are in periods, they strongly need their boyfriends to do sex with (laughs).*" Young girl, participant 10, Kayonza FGD. Regarding young boys' emotional changes during adolescence, the majority of young girls and boys perceived that when a young boy reaches puberty, his thoughts are centered around sex.

"*You know, when a boy reaches puberty, most of the time when with peers, they only talk about sexuality and about how they can entice girls of their age group to have sex*". Young boy, participant 8, Mukarange FGD.

Both young girls and boys added that some of them who fail to get friends of the opposite sex may practice masturbation and/or watch porn to learn more about sex.

**Subtheme 3: Social interaction changes.** Young girls and boys mentioned that the onset of puberty marks changes in the way adolescents feel in terms of their social interactions. They mentioned that throughout puberty, adolescents develop a need to belong to any social group among their peers. More particularly, the majority of female participants reported that when boys reach puberty, they tend to spend much of their time with girls.

**Subtheme 4: Causes of changes boys and girls experience during puberty.** Both young boys and girls viewed physical and psychological changes occurring during adolescence as a result of biological and age-related transformations. They attributed the biological causes to hormonal changes during the puberty. Boys mentioned that the hormonal changes for girls trigger the physical changes they experience and are also responsible for their menstruation. For boys, the hormonal changes cause their sperms to mature such that if they do unprotected sex with a girl in the same age group, he may impregnate her. Although in all discussion participants mentioned hormones as the major cause of the physical and psychological changes during adolescence, none of the participants mentioned the names of those hormones.

Some young boys opined that pubertal changes occur as a result of the child's living conditions and eating patterns.

"*You know, you may be from a good family and then you eat good food which can lead you to experiencing those physical changes. But, if you do not eat well, I think due to undernutrition you may not experience those physical changes later*". Young boy, participant 2, Kiziguro FGD.

Some young girls thought that changes they noticed may be a result of transition from childhood to adolescence. Some of the young girls reflected that such changes may be due to the divine powers that determine a man's growth stages.

## Theme 2: Personal experiences related to pubertal changes

Adolescents' experiences of pubertal changes are embodied into four subthemes.

**Subtheme 1: Though pubertal changes may lead to panic, they embody the attainment of maturity among young boys.** Some young boys found pubertal changes completely novel to them.

"*For me I can perhaps say that it was a bit funny. Before I became adolescent, I had no information whatsoever regarding those changes. When I first saw hair on my pubic area, I panicked and started wondering how such hair may grow on my sex area instead of my head.*" Young boy, number 3, Mulinga FGD.

Most of the male participants were surprised but excited of the changes they saw on themselves. This excitement was reported as an expressive sign of maturity.

"*I used to hear my peers saying they have grown pubic hair and beards. Whenever they were bragging about that, I got worried because I was anxious that I would not become mature soon. After some time, pubic hair and beards started growing. To me, I felt very relieved that I was becoming a man like them and that I could then look for a girlfriend.*" Young boy, participant 11, Munyiginya FGD.

**Subtheme 2: Pubertal changes affect young girls' perception of their body image.** Although there was a feeling of pride to be a mature girl by having breasts, most young girls expressed that breast budding made them feel uncomfortable and shy.

"*I remember the first time when I had my first breast buds in primary five. When I was playing with my peers, most of those I played with had not got their breast buds. They could mock me saying things like she had breasts while we do*

*not have. They said this because I was younger than them. This made me feel ashamed and uncomfortable.*" Young girl, participant 3, Kiziguro FGD.

The same feeling of shame was shared across some young girls.

"*In the early times, I was feeling shy and not comfortable. Whenever I was walking in the street even during the scorching sunshine, I was always wearing a jumper.*" Young girl, participant 4, Munyiginya FDG.

Other young girls revealed how big breasts affected their bodily image and how they affected some of the activities they enjoyed with their peers.

"*My colleagues were telling me that my breasts were big and that I had look for bras to hold them. Honestly, I was shy to tell it to my mom that I needed it. What I did, I asked a girl who was my friend, and she bought it for me.*" Young girl, participant 9, Mulinga FGD.

**Subtheme 3: Menarche causes panic and embarrassment to young girls.** Young girls experienced menarche differently but for most of the participants menarche caught them unaware and they found it embarrassing and scaring for those who had no information whatsoever and those who had little information. For all young girls, menarche though a sign of fertility and maturity, it was fearful and painful to experience. Regardless of the place and the time of the day and whether a young girl had prior information, menarche led to panic among young girls.

"*For my menarche, I was not expecting that on the day it came it should come. Even though I had some information from my elder sisters about menarche and my mummy, when I first saw blood, I got a bit scared because the way they described menarche was not how I saw it.*" Young girl, participant 5, Rwamagana FGD.

A few more other young girls revealed that they wanted their menarche would delay because, based on how their mothers or peers described menstrual bleeding, they perceived that menstruation was a nuisance to them.

"*For me, it's as if I was ready for my menarche because it happened when I was at home. Of course, I was not happy for it because of the pain I had felt. My mother used to tell me that you get some abdominal pain during periods, and I would tell her that my menarche should at least delay until I become twenty (**she laughs**).*" Young girl, participant 10, Kiziguro FGD.

Some participants reported that they had their menarche unprepared and without information. As a result, seeing blood in their genital areas became a scary thing and they reacted differently from crying to seeking emergency care to the health facility.

"*I didn't know anything about monthly bleeding. The day it happened; I was sitting on a chair watching TV with my father. When I stood up, my father asked me why I dirtied myself...I went to the washroom to check what happened to me and realized that I had blood. I cried and cried and came back telling my father to rush me to the health center immediately. When we arrived at the health facility, the nurse who received me stared at me and said, 'you are in periods.'*" Young girl, participant 1, Mulinga FGD.

**Subtheme 4: Lack of parental support for some young girls.** In Rwandan culture, discussions surrounding sexual and reproductive health are often considered taboo. Consequently, young girls have expressed that upon experiencing menarche, they felt apprehensive about disclosing this significant event to their parents fearing the uncertainties. Instead, they sought informational support from their peers.

*"When I had my menarche, I didn't tell it my parents because I was scared that they could beat me. I just secretly told one of my close friends and she gave a packet of pads and taught me how to wear a pad especially that I had no information whatsoever on how to use it.".* Young girl, participant 7, Munyiginya FGD.

## Discussion

Our research yielded five key findings: first, adolescents are aware of the physiological and bodily changes that occur in both girls and boys during puberty. However, knowledge of the emotional and psychological changes associated with puberty—such as feelings of sadness, impatience, daydreaming, depression, anxiety, changes in sleep patterns, and irritability—was notably limited. Additionally, the study revealed that adolescents often viewed pubertal changes as indicators of fertility and maturity. Despite this perception, many young girls and boys were unable to explain thoroughly the biological processes underlying the changes they experienced. Furthermore, the onset of menarche was a source of fear for some young girls, primarily due to their insufficient prior knowledge on the subject.

The study found that adolescent children in the Eastern province know the majority of the physiological and body changes a girl and a boy notice during puberty. Compared to a study conducted in Kenya and Nigeria rural areas that found that female adolescents were more likely to mention breast development than menstruation as associated with puberty [19], young boys and girls who participated exhibited a good level of understanding of all the body changes pertaining to puberty among boys and girls. This finding needs careful interpretation, as it relies on self-reported data from children. The findings of our study suggest that adolescent children demonstrate a considerable comprehension of the physical changes occurring in their bodies and the physiological effects these changes elicit, as supported by their self-reported information. However, it remains unclear whether this awareness translates into effective coping strategies for managing pubertal changes or influences their sexual behaviors. Consequently, additional research is necessary to explore and analyze the relationship between young adolescent children's awareness of pubertal changes and their sexual behaviors during the transition from puberty to adolescence. Such studies are essential for developing comprehensive interventions aimed at assisting children in managing these bodily changes, addressing misconceptions related to these changes, and helping them identify their vulnerabilities.

Compared to the body changes, our study found that both young boys and girls could not capture all emotional and mental changes associated with puberty. Participants did not mention some of the negative changes such as sorrow, impatience, daydreaming, depression, anxiety, changes in sleep pattern, and irritability that were documented from the previous studies from Kenya, Nigeria, and Iran [14,19,29]. The lack of awareness among children involved in our study regarding these adverse emotional changes can be attributed to their insufficient education on the topic both in school settings and within their families. This is particularly significant given that the impact of pubertal changes on adolescent children's mental health is a subject that has received minimal attention and exploration in the existing literature from Rwanda. Furthermore, the fact that young people from the Eastern province are not aware of these negative emotional changes has some implications on their sexuality and reproductive health. First, some of the changes may be bound up with issues relating to appearance, dress, use of cosmetics, courtship, sexuality and sexual behavior [19]. Secondly, the above-mentioned mental health issues can lead to the abuse of drugs or risky sexual behaviors. Therefore, future research needs to concentrate on the interventions that can assist adolescents to cope with these emotional and mental health changes during puberty spanning through adolescence.

Pubertal changes, particularly a boy experiencing nocturnal dreams and a girl seeing her menarche were understood as signs of fertility and maturity. This finding is consistent with other studies that established that young people take pride in the body changes occurring during adolescence [8,30]. Recognizing the physiological changes associated with fertility, adolescents may, in the absence of proper guidance, engage in sexual experimentation driven by peer pressure or a desire for self-exploration. Therefore, it is essential for schools and local communities in the Eastern Province of Rwanda

to implement regular sexual education programs, particularly during holiday periods, to equip young individuals with the knowledge necessary to navigate the complexities of pubertal changes.

We found that adolescent Children in the Eastern province of Rwanda do not know the biological phenomenon behind the changes they experience during puberty throughout adolescence. Concerted efforts are needed to enhance this critical aspect of SRH, ensuring that adolescents gain a comprehensive understanding of the biological and physiological processes associated with their developmental changes during puberty. Furthermore, it is vital to equip adolescent children with effective strategies to recognize and cope with pubertal changes positively.

Although most of female children had ever heard about menarche, at its onset some of them were scared because they had no prior knowledge about menarche. This finding has significant implications regarding the shame and stigma that girls may experience from their peers, particularly from male counterparts. It highlights the necessity for support mechanisms to help them cope with the embarrassment associated with the onset of their first menstruation. Although schools provide some level of menstrual support, there is a pressing need to enhance menstrual education and resources within communities. Our study was consistent with that from the previous studies that documented fear of menstrual blood and abdominal cramps experienced by some young girls during menstruation as worrying changes [19,31–33]. However, since the current study was not specifically exploring children's experiences and perceptions about menarche and their practices during this period, further research is needed to cover this aspect in-depth.

## Limitations

Our study has concentrated on students enrolled in twelve-year basic education secondary schools, and the results obtained cannot be generalized to encompass all children aged 10–19 in Rwanda's eastern province. Consequently, we suggest that additional research be conducted to include children of the same age attending different secondary institutions, as well as those who are not currently enrolled in school. Such studies would provide a more comprehensive understanding of how children perceive and navigate the physical and psychological transformations associated with puberty. Furthermore, we recommended that future investigations explore the connection between these pubertal changes and the mental health of children within the Rwandan context.

## Conclusion

Our research has elucidated children's comprehension of the physical and psychological transformations that occur during puberty, as well as their personal experiences of these changes. The results of this study provide a foundation for future investigations aimed at deepening the understanding of the psychological aspects of puberty, particularly in enhancing children's awareness of these changes. Such awareness is crucial for the development of effective coping strategies that promote resilience. Additionally, the findings underscore children's existing knowledge regarding pubertal changes, which can inform the design of interventions aimed at empowering children in sexual and reproductive health education, specifically addressing the challenges faced by children in the eastern province of Rwanda. Furthermore, the study identifies critical areas of focus for caregivers, enabling them to support their children in navigating the complexities of self-image and the misinformation that often accompanies pubertal changes, particularly concerning unwanted physical manifestations linked to the physiological processes of puberty. Our study suggests that adolescent students in the Eastern province still lack information about the biological phenomena leading to pubertal changes. In addition, some young girls experience menarche unprepared and get inadequate support to cope with that change in their lives. Therefore, female children's caregivers should proactively obtain menstrual pads for their daughters to ensure they are adequately prepared for school. Additionally, it is crucial to provide female students with education from qualified professionals in sexual and reproductive health. This education should focus on helping them manage the physical discomfort, emotional fluctuations, and self-esteem issues that may arise during menstruation, thereby improving their overall well-being and academic performance. Furthermore, empowering these young girls to challenge the misinformation they encounter from unreliable

sources and peers regarding the management of menstrual-related symptoms is essential. Further sexual educational interventions are needed to equip both male and female children with information on pubertal changes and coping mechanisms to adapt in order to navigate safely puberty.

## Supporting information

**S1 File. Kinyarwanda transcripts-girls.**
(DOC)

**S2 File. Kinyarwanda transcripts-boys.**
(DOC)

## Author contributions

**Conceptualization:** Kellen Muganwa, Oliva Bazirete, Marie Chantal Uwimana, Marie Grace Sandra Musabwasoni, Olive Tengera, Joy Bahumura, Hellen Nwanko Chinwe, Emmerence Uwingabire, Francoise Mujawamariya, Eugenie Mukeshimana, Richard Nsengiyumva, Christine Mbila Wabenya, Vincent Mugisha, Thierry Claudien Uhawenimana.

**Data curation:** Kellen Muganwa, Thierry Claudien Uhawenimana.

**Formal analysis:** Joy Bahumura, Emmerence Uwingabire, Elyse Muteteri, Emmanuel Habyarimana, Baudouine Ndimurukundo, Richard Nsengiyumva, Vincent Mugisha, Gerard Kaberuka, Thierry Claudien Uhawenimana.

**Funding acquisition:** Kellen Muganwa, Oliva Bazirete, Marie Chantal Uwimana, Marie Grace Sandra Musabwasoni, Joy Bahumura, Richard Nsengiyumva.

**Investigation:** Kellen Muganwa, Marie Chantal Uwimana, Emmerence Uwingabire, Francoise Mujawamariya, Emmanuel Habyarimana, Verene Abagirinana, Gerard Kaberuka, Marie Laetitia Ishimwe Bazakare.

**Methodology:** Kellen Muganwa, Oliva Bazirete, Marie Grace Sandra Musabwasoni, Olive Tengera, Joy Bahumura, Hellen Nwanko Chinwe, Emmerence Uwingabire, Elyse Muteteri, Francoise Mujawamariya, Eugenie Mukeshimana, Baudouine Ndimurukundo, Verene Abagirinana, Richard Nsengiyumva, Christine Mbila Wabenya, Gerard Kaberuka, Marie Laetitia Ishimwe Bazakare, Thierry Claudien Uhawenimana.

**Project administration:** Kellen Muganwa.

**Supervision:** Oliva Bazirete, Olive Tengera, Joy Bahumura, Hellen Nwanko Chinwe, Richard Nsengiyumva, Thierry Claudien Uhawenimana.

**Validation:** Kellen Muganwa, Gerard Kaberuka, Marie Laetitia Ishimwe Bazakare, Thierry Claudien Uhawenimana.

**Visualization:** Thierry Claudien Uhawenimana.

**Writing – original draft:** Kellen Muganwa, Oliva Bazirete, Marie Chantal Uwimana, Marie Grace Sandra Musabwasoni, Olive Tengera, Joy Bahumura, Hellen Nwanko Chinwe, Emmerence Uwingabire, Elyse Muteteri, Francoise Mujawamariya, Emmanuel Habyarimana, Eugenie Mukeshimana, Baudouine Ndimurukundo, Verene Abagirinana, Richard Nsengiyumva, Christine Mbila Wabenya, Vincent Mugisha, Gerard Kaberuka, Marie Laetitia Ishimwe Bazakare, Thierry Claudien Uhawenimana.

**Writing – review & editing:** Kellen Muganwa, Oliva Bazirete, Marie Chantal Uwimana, Marie Grace Sandra Musabwasoni, Olive Tengera, Joy Bahumura, Hellen Nwanko Chinwe, Emmerence Uwingabire, Elyse Muteteri, Francoise Mujawamariya, Emmanuel Habyarimana, Eugenie Mukeshimana, Baudouine Ndimurukundo, Verene Abagirinana, Richard Nsengiyumva, Christine Mbila Wabenya, Vincent Mugisha, Gerard Kaberuka, Marie Laetitia Ishimwe Bazakare, Thierry Claudien Uhawenimana.

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
