## [Decision Letter · Decision Letter 0]

18 Sep 2024

PONE-D-24-31307Rwandan adolescents’ awareness and experience of the pubertal changes: a qualitative studyPLOS ONE

Dear Dr. Muganwa,

Thank you for submitting your manuscript to PLOS ONE. After careful consideration, we feel that it has merit but does not fully meet PLOS ONE’s publication criteria as it currently stands. Therefore, we invite you to submit a revised version of the manuscript that addresses the points raised during the review process.

We look forward to receiving your revised manuscript.

Kind regards,

Shadab Shahali, PHD

Academic Editor

PLOS ONE

Additional Editor Comments:

The introduction part in abstract contains grammatical errors and awkward phrasing, making it difficult to understand. For example, “Adolescence experienced puberty changes both biological and psychological changes that takes place during adolescent age and affects them differently” is confusing and should be rephrased for clarity.

Its not highlight what is novel or unique about this study compared to previous research on the topic. Emphasizing any new contributions or insights would strengthen the abstract.

Improved sentence structure and grammar for better readability and understanding.

Ensured consistent use of terms such as “puberty,” “adolescents,” and “children.”

The methodology mentions a descriptive qualitative design but does not provide details on the specific qualitative methods used

The sampling approach is not clearly justified. The criteria for selecting participants are not fully explained.

The discussion reiterates the findings but does not delve deeply into their implications.

While the discussion compares the study’s findings with those from Kenya and Nigeria, it does not provide a detailed analysis of why these differences might exist. Exploring cultural, social, or educational factors that could explain these differences would add depth.

The conclusion reiterates the need for future research but does not provide specific recommendations or actionable steps based on the study’s findings.

The conclusion does not emphasize the significance or potential impact of the study’s findings on the field of adolescent health or education.

Reviewers' comments:

Reviewer's Responses to Questions

**Comments to the Author**

1. Is the manuscript technically sound, and do the data support the conclusions?

Reviewer #1: Partly

2. Has the statistical analysis been performed appropriately and rigorously? 

Reviewer #1: N/A

3. Have the authors made all data underlying the findings in their manuscript fully available?

Reviewer #1: Yes

4. Is the manuscript presented in an intelligible fashion and written in standard English?

Reviewer #1: No

5. Review Comments to the Author

Reviewer #1: The introduction part in abstract contains grammatical errors and awkward phrasing, making it difficult to understand. For example, “Adolescence experienced puberty changes both biological and psychological changes that takes place during adolescent age and affects them differently” is confusing and should be rephrased for clarity.

Its not highlight what is novel or unique about this study compared to previous research on the topic. Emphasizing any new contributions or insights would strengthen the abstract.

Improved sentence structure and grammar for better readability and understanding.

Ensured consistent use of terms such as “puberty,” “adolescents,” and “children.”

The methodology mentions a descriptive qualitative design but does not provide details on the specific qualitative methods used

The sampling approach is not clearly justified. The criteria for selecting participants are not fully explained.

The discussion reiterates the findings but does not delve deeply into their implications.

While the discussion compares the study’s findings with those from Kenya and Nigeria, it does not provide a detailed analysis of why these differences might exist. Exploring cultural, social, or educational factors that could explain these differences would add depth.

The conclusion reiterates the need for future research but does not provide specific recommendations or actionable steps based on the study’s findings.

The conclusion does not emphasize the significance or potential impact of the study’s findings on the field of adolescent health or education.

6. PLOS authors have the option to publish the peer review history of their article (what does this mean? ). If published, this will include your full peer review and any attached files.

**Do you want your identity to be public for this peer review?** For information about this choice, including consent withdrawal, please see our Privacy Policy .

Reviewer #1: No

---

## [Author Response · Author response to Decision Letter 1]

25 Oct 2024

Comments Responses Page within the manuscript

Thank you for the comment. We have checked the requirements and have adhered to them.

The introduction part in abstract contains grammatical errors and awkward phrasing, making it difficult to understand. For example, “Adolescence experienced puberty changes both biological and psychological changes that takes place during adolescent age and affects them differently” is confusing and should be rephrased for clarity. Thank you for the comment. We have revised the introductory part of the abstract and addressed the grammatical errors and awkward phrasing that were appearing.

See Page 1

Its not highlight what is novel or unique about this study compared to previous research on the topic. Emphasizing any new contributions or insights would strengthen the abstract. We really appreciate this comment. First, we revised the conclusion of the manuscript to indicate what our study is contributing to the evidence and practice. In addition, in the main manuscript we have revised the discussion and conclusion sections to highlight the uniqueness of this study.

Improved sentence structure and grammar for better readability and understanding. We have crosschecked the manuscript to address some grammatical errors and structure related issues that affected the readability of the manuscript. Changes have been made throughout the manuscript.

Ensured consistent use of terms such as “puberty,” “adolescents,” and “children.” Thank you for these important observations. We resolved to use puberty and children to ensure consistency

See pages 1,2,3,5,6,7, 10,12,16,19,20,21 and 22

The methodology mentions a descriptive qualitative design but does not provide details on the specific qualitative methods used The comment is appreciated. Details were given on the specific qualitative methods used. 7

The sampling approach is not clearly justified. The criteria for selecting participants are not fully explained. The sampling approach were clearly justified and selecting participants criteria were explained 7-8

The discussion reiterates the findings but does not delve deeply into their implications Thank you for this important observation. We improved the discussion section to also comment on the implications of some of the findings.

19-22

While the discussion compares the study’s findings with those from Kenya and Nigeria, it does not provide a detailed analysis of why these differences might exist. Exploring cultural, social, or educational factors that could explain these differences would add depth Thanks for the comment. A detailed analysis on comparison study findings were addressed in the manuscript. 20-21

The conclusion does not emphasize the significance or potential impact of the study’s findings on the field of adolescent health or education Thank you very much. We revised and elaborated the conclusion section of our manuscript to indicate the significance or potential impact of our study’s findings on the field of adolescent children’s health or education. 22-23

---

## [Decision Letter · Decision Letter 1]

2 Jan 2025

PONE-D-24-31307R1Rwandan adolescent children’s awareness and experience of the pubertal changes: a qualitative studyPLOS ONE

Dear Dr. Muganwa,

Thank you for submitting your manuscript to PLOS ONE. After careful consideration, we feel that it has merit but does not fully meet PLOS ONE’s publication criteria as it currently stands. Therefore, we invite you to submit a revised version of the manuscript that addresses the points raised during the review process.

We look forward to receiving your revised manuscript.

Kind regards,

Shadab Shahali, PHD

Academic Editor

PLOS ONE

**Additional Editor Comments:**

I have read your article and hope to be able to give you a final decision on the article after receiving these revisions.

Reviewers' comments:

Reviewer's Responses to Questions

**Comments to the Author**

1. If the authors have adequately addressed your comments raised in a previous round of review and you feel that this manuscript is now acceptable for publication, you may indicate that here to bypass the “Comments to the Author” section, enter your conflict of interest statement in the “Confidential to Editor” section, and submit your "Accept" recommendation.

Reviewer #1: (No Response)

Reviewer #2: (No Response)

2. Is the manuscript technically sound, and do the data support the conclusions?

Reviewer #1: (No Response)

Reviewer #2: No

3. Has the statistical analysis been performed appropriately and rigorously? 

Reviewer #1: (No Response)

Reviewer #2: N/A

4. Have the authors made all data underlying the findings in their manuscript fully available?

Reviewer #1: (No Response)

Reviewer #2: Yes

5. Is the manuscript presented in an intelligible fashion and written in standard English?

Reviewer #1: (No Response)

Reviewer #2: No

6. Review Comments to the Author

Reviewer #1: Please provide a table or image illustrating the process of extracting themes from the code.

Considering that the immersion of a qualitative researcher is necessary for extracting data, how was this done by a team, please mention the evidence of this method.

How is the trustworthiness of the study measured?

Reviewer #2: Your article is valuable, but it has fundamental flaws, which are listed below.

Title:

1-Rwandan may be unknown to many readers from other continents. Maybe it would be better to write about African adolescents.

2-In title: adolescent or children??? The title should indicate the target group.

3- Rwandan adolescent children’s awareness and experience of the pubertal changes: a qualitative study:

suggest a change to :

Adolescent awareness and experience of the pubertal changes: a qualitative study from Africa

Abstract:

1-Modify the abstract structure according to the journal guidelines. It is too long and vague.

2-Mention the purpose of the study in the abstract.

3-Mention the method of your analysis in the abstract.

4-You wrote that 4 themes were identified. Write the names of the themes and categories in the findings.

5-Please, rewrite the conclusion based on the results.

6- Choose our keywords based on mesh

Introduction:

1-The manuscript needs editing as it lacks proficiency in English.

2-Long paragraphs with only one reference??

3-The introduction is too long and vague, does not indicate the purpose of the study, does not indicate the GAP knowledge

4- You must demonstrate the innovation of your work in the introduction.

Methods:

1-There is no need to define adolescence based on WHO in the method.

2-What are your inclusion and exclusion criteria?

3-Did you use any special software to analyze qualitative data?

4-Explain more about the analysis method.

5-Mention some of the interview questions.

6-On what basis were the interview questions developed?

7-Was there a specific order in which the questions were asked in the interview?

Results and discussion:

1-You wrote 4 themes in the abstract, and 5 themes in the findings. Which one is correct??

2-Theme names should be short, descriptive, and comprehensive.

3-Please provide one sentence from the contributors for each sub-category.

4-The findings are too long. It is suggested that you present the findings in a table and briefly summarize them in the text.

5-You just had a theme?

How about sub-themes or codes?

6-In the first paragraph, briefly present the findings and in the following paragraphs, discuss the findings.

7-what are your limitations, strengths points, and suggestions?

7. PLOS authors have the option to publish the peer review history of their article (what does this mean? ). If published, this will include your full peer review and any attached files.

**Do you want your identity to be public for this peer review?** For information about this choice, including consent withdrawal, please see our Privacy Policy .

Reviewer #1: No

Reviewer #2: No

---

## [Author Response · Author response to Decision Letter 2]

14 Feb 2025

Point-by-point response to reviewers’ comments

Comments How the comments have been addressed

Reviewer 1

Please provide a table or image illustrating the process of extracting themes from the code.

Considering that the immersion of a qualitative researcher is necessary for extracting data, how was this done by a team, please mention the evidence of this method. Thank you for this insight. We have provided a table illustrating how themes and subthemes emerged.

How is the trustworthiness of the study measured? Thank you very much. Since this paper is qualitative, we have indicated how we followed rigorous methods to reach transferable findings.

Reviewer 2

Your article is valuable, but it has fundamental flaws, which are listed below. Thank you for the feedback. We have tried as much as we could to address some of the flaws you have highlighted as you will find in the revised version of this manuscript.

1-Rwandan may be unknown to many readers from other continents. Maybe it would be better to write about African adolescents. We greatly appreciate your valuable insight. After careful consideration of your suggestion, we concluded that it is essential to retain 'Rwanda' in the title. This decision is based on the fact that we have not conducted a similar study in any other African country, which would provide a broader comparative context beyond Rwanda.

2-In title: adolescent or children??? The title should indicate the target group. We maintained adolescent as the target group.

3- Rwandan adolescent children’s awareness and experience of the pubertal changes: a qualitative study:

suggest a change to :

Adolescent awareness and experience of the pubertal changes: a qualitative study from Africa We have formulated the study title as follows: ‘Adolescent awareness and experience of the pubertal changes: a qualitative study from Rwanda’

Abstract:

1-Modify the abstract structure according to the journal guidelines. It is too long and vague. We have revised the abstract section to make it clear and made it succinct from 417 words to 276 words.

2-Mention the purpose of the study in the abstract. Addressed

3-Mention the method of your analysis in the abstract. Thank you. We have addressed this.

4-You wrote that 4 themes were identified. Write the names of the themes and categories in the findings. Thank you. We have addressed the comment.

5-Please, rewrite the conclusion based on the results. The conclusion was rewritten to reflect the results.

6- Choose our keywords based on mesh Addressed.

Introduction:

1-The manuscript needs editing as it lacks proficiency in English. We appreciate the comment, and we have particularly paid attention to reducing the issue of long paragraph highlighted. Though English is not our first language, we have done our best to make the section more comprehensible.

2-Long paragraphs with only one reference?? Thank you for highlighting this issue. We have revised the introduction section to reduce the length of paragraphs and further improve the references.

3-The introduction is too long and vague, does not indicate the purpose of the study, does not indicate the GAP knowledge We appreciate this observation. Based on this, we have revised the introduction section to first reduce its length, demonstrate the research gap, and further highlighted why this study was needed in the context of Rwanda.

4- You must demonstrate the innovation of your work in the introduction. Thank you for the comment. Our study did not intend to develop any intervention as a innovation but instead we sought to first highlight gaps in adolescents’ knowledge of pubertal changes and from their experiences of puberty, we sought to highlight areas that would need strengthening in the current sexual education programs in Rwanda.

Methods:

1-There is no need to define adolescence based on WHO in the method. Thank you. We have addressed this issue.

2-What are your inclusion and exclusion criteria? Thank you. We have added the exclusion criteria to complement the inclusion criteria for our study.

3-Did you use any special software to analyze qualitative data? Thank you. We used Dedoose software to analyse the data.

4-Explain more about the analysis method. Thank you very much. We have revised the sub-section of data analysis to explain the process of analyzing our data.

5-Mention some of the interview questions. We have provided the key questions that comprised the topic guide used during the moderation of the focus group discussions.

6-On what basis were the interview questions developed? Thank you. We have indicated that the questions were developed based on the study’s objectives and literature about adolescents’ experiences of pubertal changes.

7-Was there a specific order in which the questions were asked in the interview? We appreciate your feedback. In the manuscript, we have provided a list of questions posed to participants, and the sequence in which they are presented mirrors the order utilized by researchers to facilitate the focus group discussion.

Results and discussion:

1-You wrote 4 themes in the abstract, and 5 themes in the findings. Which one is correct?? Thank you for your observation. With the current revised version, two major themes and their subthemes are presented.

2-Theme names should be short, descriptive, and comprehensive Thank you. Within the current version, we took into consideration this comment. Theme names have been shortened to ensure they reflect the study objectives. In addition, we also subdivided the themes into sub-themes reflecting themes to ensure more comprehension.

3-Please provide one sentence from the contributors for each sub-category. Thank you. We have considered this suggestion.

4-The findings are too long. It is suggested that you present the findings in a table and briefly summarize them in the text. We really appreciate this observation. We have summarized tried to make the results section succinct and more comprehensive as suggested.

5-You just had a theme? How about sub-themes or codes? Thank you. We have taken the comment into consideration.

6-In the first paragraph, briefly present the findings and in the following paragraphs, discuss the findings. Thank you for your observation. We have addressed this in the revised manuscript. See page…

7-what are your limitations, strengths points, and suggestions? Thank you for this suggestion. Throughout the discussion section we have highlighted some limitations and suggestions for further work to expand on our findings. We have also provided a section of methodology limitation of our study and suggested further research.

---

## [Decision Letter · Decision Letter 2]

1 Apr 2025

PONE-D-24-31307R2Adolescent awareness and experience of the pubertal changes: a qualitative study from Rwanda’PLOS ONE

Dear Dr. Muganwa,

Thank you for submitting your manuscript to PLOS ONE. After careful consideration, we feel that it has merit but does not fully meet PLOS ONE’s publication criteria as it currently stands. Therefore, we invite you to submit a revised version of the manuscript that addresses the points raised during the review process.

We look forward to receiving your revised manuscript.

Kind regards,

Shadab Shahali, PHD

Academic Editor

PLOS ONE

Journal Requirements:

Reviewers' comments:

Reviewer's Responses to Questions

**Comments to the Author**

1. If the authors have adequately addressed your comments raised in a previous round of review and you feel that this manuscript is now acceptable for publication, you may indicate that here to bypass the “Comments to the Author” section, enter your conflict of interest statement in the “Confidential to Editor” section, and submit your "Accept" recommendation.

Reviewer #2: (No Response)

2. Is the manuscript technically sound, and do the data support the conclusions?

Reviewer #2: Partly

3. Has the statistical analysis been performed appropriately and rigorously? 

Reviewer #2: I Don't Know

4. Have the authors made all data underlying the findings in their manuscript fully available?

Reviewer #2: Yes

5. Is the manuscript presented in an intelligible fashion and written in standard English?

Reviewer #2: No

6. Review Comments to the Author

Reviewer #2: The authors have responded to a number of comments.

But I still think it is poor in terms of writing and English.

The interview questions are not clearly stated.

There are inaccuracies in the text, for example, the themes changed after revision. But the names of the themes in the text and the abstract are not the same.

Innovation and creativity in a study have nothing to do with intervention. Every study should have creativity. You didn't mention innovation and the gap knowledge.

The inclusion and exclusion criteria are not modified. The inclusion criteria should mention all the characteristics of the participants. You should know that the readers should fully understand your study. For example, gender is not mentioned in the inclusion criteria.

7. PLOS authors have the option to publish the peer review history of their article (what does this mean? ). If published, this will include your full peer review and any attached files.

**Do you want your identity to be public for this peer review?** For information about this choice, including consent withdrawal, please see our Privacy Policy .

Reviewer #2: No

---

## [Author Response · Author response to Decision Letter 3]

25 Apr 2025

Comments How the comments have been addressed

Academic Editor

Please review your reference list to ensure that it is complete and correct. If you have cited papers that have been retracted, please include the rationale for doing so in the manuscript text, or remove these references and replace them with relevant current references. Any changes to the reference list should be mentioned in the rebuttal letter that accompanies your revised manuscript. If you need to cite a retracted article, indicate the article’s retracted status in the References list and also include a citation and full reference for the retraction notice. We appreciate your guidance. Upon reviewing the references, we found that none of the studies we cited have been retracted.

Reviewer 2

The authors have responded to a number of comments.

But I still think it is poor in terms of writing and English. We appreciate your feedback. We have thoroughly reviewed the manuscript and made corrections to several issues, specifically where words were omitted, and we have also improved the sentence structure in various sections to enhance the overall readability of the paper.

The interview questions are not clearly stated. Thank you very much. Since we have included the interview guide in the supplementary documents, we deemed it unnecessary to incorporate all the questions from the main manuscript.

There are inaccuracies in the text, for example, the themes changed after revision. But the names of the themes in the text and the abstract are not the same. We have examined the text and corrected the inaccuracies. The content in the results section of the abstract has been aligned to correspond with the findings obtained.

Innovation and creativity in a study have nothing to do with intervention. Every study should have creativity. You didn't mention innovation and the gap knowledge. Thank you very much for this observation. We have carefully considered this issue and highlighted the gap in knowledge that led to the conduct of this study. For details, see pages 5-6 of the revised document with track changes.

The inclusion and exclusion criteria are not modified. The inclusion criteria should mention all the characteristics of the participants. You should know that the readers should fully understand your study. For example, gender is not mentioned in the inclusion criteria. Thank you very much. We have revised the inclusion and exclusion criteria. Clarifications were made to show who was included and excluded from the study. For detail, see page 7 of the revised document with track changes.

---

## [Decision Letter · Decision Letter 3]

14 May 2025

Adolescent awareness and experience of the pubertal changes: a qualitative study from Rwanda’

PONE-D-24-31307R3

Dear Dr.Kellen Muganwa

We’re pleased to inform you that your manuscript has been judged scientifically suitable for publication and will be formally accepted for publication once it meets all outstanding technical requirements.

Kind regards,

Shadab Shahali, PHD

Academic Editor

PLOS ONE

Additional Editor Comments (optional):

Reviewers' comments:

Reviewer's Responses to Questions

**Comments to the Author**

1. If the authors have adequately addressed your comments raised in a previous round of review and you feel that this manuscript is now acceptable for publication, you may indicate that here to bypass the “Comments to the Author” section, enter your conflict of interest statement in the “Confidential to Editor” section, and submit your "Accept" recommendation.

Reviewer #1: (No Response)

2. Is the manuscript technically sound, and do the data support the conclusions?

Reviewer #1: (No Response)

3. Has the statistical analysis been performed appropriately and rigorously? 

Reviewer #1: (No Response)

4. Have the authors made all data underlying the findings in their manuscript fully available?

Reviewer #1: (No Response)

5. Is the manuscript presented in an intelligible fashion and written in standard English?

Reviewer #1: (No Response)

6. Review Comments to the Author

Reviewer #1: (No Response)

7. PLOS authors have the option to publish the peer review history of their article (what does this mean? ). If published, this will include your full peer review and any attached files.

**Do you want your identity to be public for this peer review?** For information about this choice, including consent withdrawal, please see our Privacy Policy .

Reviewer #1: **Yes: ** Shadab Shahali

---

## [Editor Report · Acceptance letter]

PONE-D-24-31307R3

PLOS ONE

Dear Dr. Muganwa,

I'm pleased to inform you that your manuscript has been deemed suitable for publication in PLOS ONE. Congratulations! Your manuscript is now being handed over to our production team.

Kind regards,

on behalf of

Dr. Shadab Shahali

Academic Editor

PLOS ONE